# Exploring Segment Anything Foundation Models for Out of Domain Crevasse Drone Image Segmentation

Steven Wallace*[1,3], Aiden Durrant[1], William D. Harcourt[2,3], Richard Hann[4], and Georgios Leontidis[1,3]

[1]School of Natural and Computing Sciences, University of Aberdeen, UK
[2]School of Geosciences, University of Aberdeen, UK
[3]Interdisciplinary Institute, University of Aberdeen, UK
[4]Department of Engineering Cybernetics, Norwegian University of Science and Technology, Norway

## Abstract

In this paper, we explore the application of Segment Anything (SAM) foundation models for segmenting crevasses in Uncrewed Aerial Vehicle (UAV) images of glaciers. We evaluate the performance of the SAM and SAM 2 models on ten high-resolution UAV images from Svalbard, Norway. Each SAM model has been evaluated in inference mode without additional fine-tuning. Using both automated and manual prompting methods, we compare the segmentation quantitatively using Dice Score Coefficient (DSC) and Intersection over Union (IoU) metrics. Results show that the SAM 2 Hiera-L model outperforms other variants, achieving average DSC and IoU scores of 0.43 and 0.28 respectively with automated prompting. However, the overall off-the-shelf performance suggests that further improvements are still required to enable glaciologists to examine crevasse patterns and associated physical processes (e.g. iceberg calving), indicating the need for further fine-tuning to address domain shift challenges. Our results highlight the potential of segmentation foundation models for specialised remote sensing applications while also identifying limitations in applying them to high-resolution UAV images, as well as ways to enhance further model performance on out-of-domain glacier imagery, such as few-shot and weakly supervised learning techniques.

## 1 Introduction

In recent years, deep learning foundation models that contain the transformer neural network architecture have gained significant traction in natural language processing and computer vision [1, 2]. Notably, large companies like Meta Facebook AI Research (FAIR) and Google are contributing to the evolution of deep learning research by publishing more pre-trained models open source [3–5]. Convolutional Neural Networks (CNNs) and the Vision Transformer (ViT) have advanced the field by providing researchers with access to many pre-trained models that would otherwise be computationally prohibitive to pre-train

from scratch on non-commercial computer hardware [6, 7]. In most cases, pre-training on a larger dataset in a controlled laboratory with more compute power and then fine-tuning to a specialist downstream task usually leads to better performance at a much cheaper cost [8]. However, foundation models still face challenges in generalising to specialised image segmentation applications from the Earth Observation (EO) and medical image domains because of the large quantities of labelled data required for conventional supervised fine-tuning [9, 10]. Gathering and labelling image segmentation data can be labour-intensive, requires expert approval, and is costly when acquiring data in the Polar regions due to their remote locality [11, 12]. Therefore, the Segment Anything Model (SAM) and the recently released Segment Anything Model 2 (SAM 2) from Meta FAIR provide researchers with the first segmentation foundation models [13, 14]. SAM and SAM 2 are pre-trained on large-scale image datasets with self-supervised learning but their applicability and performance on more specialist downstream applications for the segmentation of fractures on glaciers known as crevasses, is underexplored. This work explores the performance and generalisability of the SAM and SAM 2 foundation models when applied to centimetre resolution UAV images captured over a fast-flowing glacier in Svalbard using automatic and manual prompting without additional fine-tuning. The SAM and SAM 2 image segmentation architecture is included in Figure 1.

### 1.1 Motivation and Challenges

Melting and retreating glaciers worldwide have garnered significant interest as they directly contribute to rising sea levels [15]. Climate change and rising sea levels are causing concern around the world because of the disruption and devastating effects on coastal habitats [16]. In this context, understanding better the controls on glacier mass balance, which directly measures the amount of ice being lost, will help scientists analyse glacier dynamics in more detail. In the Arctic, which has been warming four times faster than anywhere else on the globe since 1979, Svalbard has lost ice cover at a rate of -10.5

*Corresponding Author: s.wallace.23@abdn.ac.uk

Proceedings of the 6th Northern Lights Deep Learning Conference (NLDL), PMLR 265, 2025.

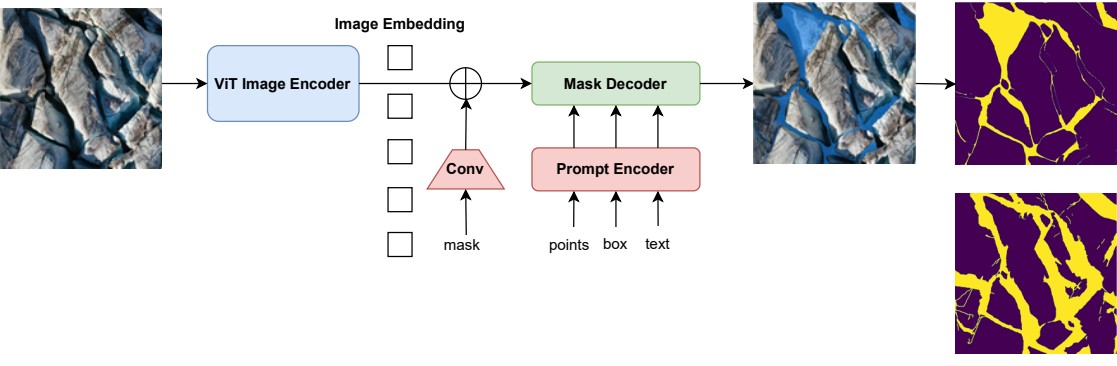

**Figure 1.** The SAM and SAM 2 architecture that is used for image segmentation. An input UAV image is included with the output segmentation mask over the top of the image as the output. The binary segmentation mask and the ground truth have also been included in the diagram.

Gt / yr in the early 21st century [17, 18]. According to [15] a total of 7563 ± 699 Gt of ice has been lost from ice sheets between 1992 and 2020.

Crevasses on the surface of glaciers provide the line of weakness through which icebergs detach from tidewater glacier termini in a process known as iceberg calving [19, 20]. High rates of iceberg calving lead to glacier mass loss, hence it follows that increased crevassing may lead to further calving and hence mass loss. Furthermore, crevasses also play an essential role in regulating ice flow by acting as a conduit for surface meltwater to reach the bed and speed up ice flow [19]. However, manually mapping crevasses with Geographic Information System (GIS) software is laborious and time-consuming because of the the high level of skill required to accurately annotate crevasses. Therefore, automatically annotating crevasses using deep learning foundation models provides a solution to a labour-intensive process. However, like other remote sensing applications such as building or sea ice detection, crevasse segmentation usually involves complex scenes not found in everyday images, making it harder for algorithms to segment [21].

Using a foundation model to prompt an image and output a segmentation mask could help with starting the crevasse mapping process for a glacier or provide enough detail to give experts enough information to discover new findings. Currently, to date, a segmentation model that requires or does not require any prompting to segment crevasses to this standard is not available off the shelf. Therefore, the deep learning community and glaciologists in particular, could use the SAM or SAM 2 models built into an end-to-end user interface system to start the labelling process for segmentation to reduce labelling time and the amount of labour required.

The contributions of this paper are summarised as follows. UAV images that are an optical image data modality have been compared in performance using the SAM and SAM 2 foundation models for image segmentation. The binary segmentation mask output from the SAM and SAM 2 models have been compared visually and with evaluation performance metrics for image segmentation applications. We also provide suggestions for future work that could improve the output segmentation results from the SAM 2 foundation model in the form of few-shot or weakly supervised learning.

## 2 Related Work

### 2.1 Glaciology Deep Learning

In recent years, automated algorithms have been explored because of how labour-intensive it is to manually map crevasses using GIS software [23]. The U-Net model was initially used to extract crevasses from the entire Antarctic Ice Shelf using Mean of Angles (MOA) images, demonstrating the use of deep learning for crevasse extraction [24, 25]. In [26] the U-Net model was enhanced by pre-processing the input Sentinel-1 Synthetic Aperture Radar (SAR) images from Antarctica with the Probabilistic-Patch Based filter to highlight crevasses in the SAR imagery. Further data augmentations were applied to increase the size of the dataset to train the U-Net model [26]. A Dice Score Coefficient (DSC) of 0.7602 was achieved with this method. However, the U-Net model failed to segment crevasses on the glacier where ridges or raised land passes underneath the ice, creating curvature on the surface. A further two models were proposed in [23] to distinguish between crevasses on Ice Shelves and Ice Sheets from Antarctica with the help of the proposed Parallel-Structure filter. Two types of crevasses had to be distinguished with each of the two models because of the angle of the radar signal often not capturing finer crevasses that are only a single pixel wide [23]. Therefore,

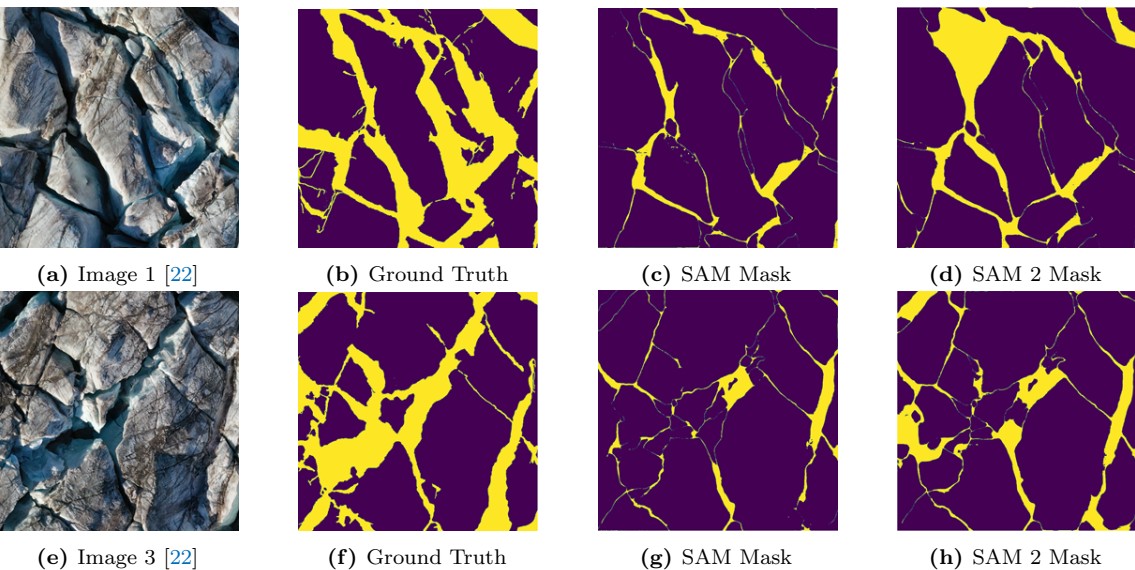

**(a)** Image 1 [22]  **(b)** Ground Truth  **(c)** SAM Mask  **(d)** SAM 2 Mask

**(e)** Image 3 [22]  **(f)** Ground Truth  **(g)** SAM Mask  **(h)** SAM 2 Mask

**Figure 2.** Example UAV drone images, segmentation masks, and ground truth masks from the best-performing images segmented by the SAM and SAM 2 models, respectively, using the mask generator module. The segmentation mask output for the SAM and SAM 2 models are included for both images.

thinner crevasses were eliminated to allow the overall algorithm to concentrate on larger crevasses that are often involved with glacier calving. The ResUNet model that adds residual connections to a standard U-Net architecture was proposed by [27] to further enhance the performance of crevasse identification where a DSC of 0.771 was achieved over 0.751 when using a standard U-Net architecture. In [20] MobileViT was used as a lightweight backbone feature extractor fine-tuned on SAR imagery from the West Antarctic Ice Sheet (WAIS). The backbone ViT was pre-trained on ImageNet-21K and fine-tuned on ImageNet-1K to increase the size of the pre-training datasets [28, 29]. Pre-trained backbone models trained on well-constructed datasets such as ImageNet as a feature extractor allow deep learning segmentation models to be adapted using less data than training the overall model from scratch. A DSC of 0.840 was achieved, improving the performance of Antarctic crevasse detection without any pre-processing steps for the input images. All of the above techniques have been applied to Antarctic crevasse segmentation rather than in the Arctic. The reasons for this are that crevasses in Antarctica are larger and appear on satellite imagery that is openly available for machine/deep learning research. As crevasses in the Arctic are smaller and harder to detect than in Antarctica, they are more challenging to segment for computerised algorithms.

## 2.2 Segment Anything Model (SAM)

The SAM and SAM 2 foundation models released in 2023 and 2024 respectively by Meta FAIR are the first foundation models for image and video

segmentation [13, 14]. The SAM models provided researchers with an open-source foundation model to use for segmentation applications with prompting. The main differences between the SAM models are that the SAM 2 model has smaller, more efficient ViT backbone models that are based on the Masked Auto Encoder Hiera (MAE-Hiera) models [30], SAM 2 can be used on image and video data and the overall SAM 2 model has been trained on the SA-1B and SA-V datasets [13, 14]. In total, the SAM 2 model has been trained on one billion patches from eleven million images from the SA-1B dataset and 35.5 million video frames as image patches from the SA-V dataset. Overall, this is a total of 1.355 billion image patches. Therefore, the SAM 2 model is not only more lightweight in architecture but is the segmentation foundation model that has been trained on the largest quantity of image data available to date. However, as SAM models have recorded excellent performance metrics on general AI benchmark datasets that are in a similar domain to the training data of the SA-1B and SA-V datasets, it is challenging to adapt them to complex imagery such as for EO or medical applications. In [31, 32] SAM was used to segment glaciological features across the EO platforms from Sentinel 1, Sentinel 2, and Planet. A DSC value of 0.44 was achieved with manual point prompts without any additional fine-tuning for crevasse segmentation on a Planet optical image from the Helheim glacier in East Greenland. Being able to prompt a foundation model to an accurate output prediction for crevasse segmentation would help overcome data labelling challenges that are associated with mapping crevasse patterns.

| Metric | SAM-B | SAM-L | SAM-H | SAM2-T | SAM2-S | SAM2-B+ | SAM2-L |
|--------|-------|-------|-------|--------|--------|---------|--------|
| DSC | 0.37 | 0.36 | 0.34 | 0.42 | 0.40 | 0.40 | **0.43** |
| IoU | 0.24 | 0.22 | 0.21 | 0.27 | 0.26 | 0.26 | **0.28** |

**Table 1.** SAM and SAM 2 mask generator segmentation results (average DSC and IoU scores) of seven SAM model variants on ten 2D RGB UAV images from the Borebreen glacier. SAM-Base: 93.7M; SAM-Large: 312.3M; SAM-Huge: 641.1M; SAM2-Tiny: 38.9M; SAM2-Small: 46.0M; SAM2-Base+: 80.8M; SAM2-Large: 224.4M.

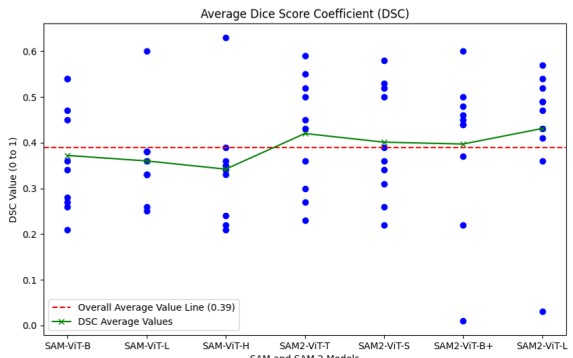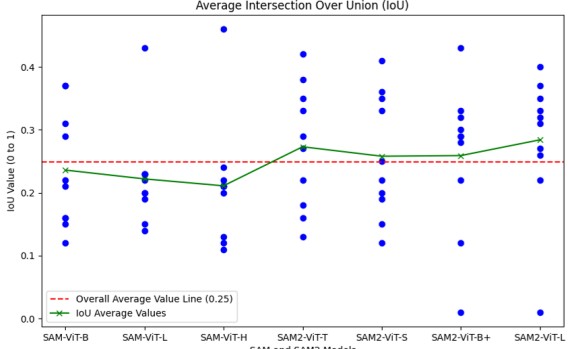

**Figure 3.** DSC and IoU dust line plots for the SAM and SAM 2 results on all seven models for the automatic mask generator prompt experiments on ten 2D RGB UAV images from the Borebreen glacier.

## 3 UAV Data

The spatial resolution of satellite imagery is much coarser compared to those acquired from UAVs, enabling mapping of finer scale structures on a glacier surface [33]. In this study, we used UAV imagery collected over the highly crevassed glacier terminus region in Svalbard [22]. The images were extracted with a variety of crevasse patterns to test the performance of the SAM models in a range of conditions. Structure from Motion (SfM) was used to construct an orthomosaic of the surveyed region at 10 cm pixel spacing from images captured on 8th August 2023 [34]. Figure 2 displays an example of the best UAV images segmented from the SAM and SAM 2 models that were captured from the Borebreen glacier in Svalbard, Norway.

The UAV images were captured on 8th August 2023 because Svalbard in the Arctic is in the summer season. Capturing UAV image data in the summer months allows for maximum use of 24-hour daylight conditions in the Arctic as opposed to permanent darkness in the winter. Permanent darkness would prevent optical imagery from being captured because optical image sensors require ultraviolet (UV) rays from the sun to operate. UAV images are better in resolution than satellite imagery captured from space because UAVs fly 20 meters above the ground. Another limitation when using optical satellite imagery from space is that if there is any cloud cover or snow on the ground, it is captured with the image. Therefore, these limitations are removed by using UAV optical image data during summer because they fly below the cloud level, and there is less snow on the ground in the Arctic. The resolution

at 10 cm per pixel in the UAV images provides a better resolution than 3 to 10 meters per pixel from satellite images captured from space. Therefore, the crevasses are larger in UAV imagery, making them less challenging to segment for computerised algorithms.

The UAV images used for the experiments were cropped from the larger orthomosaic image in the GIS software package QGIS. The output image crops were converted from GeoTiff to RGB format and cropped to a resolution of 1024 x 1024 pixels each using the RasterIO package available to Python before modelling. An image resolution of 1024 x 1024 has been used to prevent images from being interpolated up to a larger resolution that is required to align with the input dimensions of the SAM model. Therefore, as much information in the image can be retained to improve the performance of the SAM model and allow a fairer evaluation against SAM 2. In total, ten images were converted to RGB format and cropped to allow them to be manually annotated for segmentation in the Computer Vision Annotation Tool (CVAT). Each of the ten annotated images was approved by an expert glaciologist to ensure proper evaluation of any modelling.

## 4 Methods

### 4.1 Mask Generator Experiments

The SAM and SAM 2 model experiments have been carried out in the same way by using the automated mask generator and manual point prompt Python source code provided in the official GitHub repository for each model [35, 36]. Using the mask gen-

| Metric | SAM-B | SAM-L | SAM-H | SAM2-T | SAM2-S | SAM2-B+ | SAM2-L |
|--------|-------|-------|-------|--------|--------|---------|--------|
| DSC | 0.09 | 0.11 | **0.15** | 0.11 | 0.09 | 0.08 | 0.08 |
| IoU | 0.05 | 0.06 | **0.08** | 0.06 | 0.05 | 0.06 | 0.05 |

**Table 2.** SAM and SAM 2 single-mask point-prompt segmentation results (average DSC and IoU scores) of seven SAM model variants on ten 2D RGB UAV images from the Borebreen glacier. SAM-Base: 93.7M; SAM-Large: 312.3M; SAM-Huge: 641.1M; SAM2-Tiny: 38.9M; SAM2-Small: 46.0M; SAM2-Base+: 80.8M; SAM2-Large: 224.4M.

| Metric | SAM1-B | SAM1-L | SAM1-H | SAM2-T | SAM2-S | SAM2-B+ | SAM2-L |
|--------|--------|--------|--------|--------|--------|---------|--------|
| DSC | 0.16 | **0.21** | 0.20 | 0.19 | 0.18 | 0.12 | 0.19 |
| IoU | 0.09 | **0.12** | **0.12** | 0.11 | 0.11 | 0.07 | **0.12** |

**Table 3.** SAM and SAM 2 best multi-mask point-prompt segmentation results (average DSC and IoU scores) of seven SAM model variants on ten 2D RGB UAV images from the Borebreen glacier. SAM-Base: 93.7M; SAM-Large: 312.3M; SAM-Huge: 641.1M; SAM2-Tiny: 38.9M; SAM2-Small: 46.0M; SAM2-Base+: 80.8M; SAM2-Large: 224.4M.

erator to generate prompts on the UAV images automatically is a way to prompt the SAM and SAM 2 models without a human placing point or box prompts on the input image. Therefore, making it faster to apply prompts to each image tested during the evaluation of any modelling.

The SAM and SAM 2 models are designed to segment objects in images [13, 14]. Therefore, as the ice on the glacier in the foreground is larger than the crevasses, the ice is prompted by the mask generator module. The mask generator from SAM and SAM 2 outputs a binary segmentation mask for each object in an image as an instance. Therefore, each segmentation mask for each instance must be combined into one overall binary segmentation mask in preparation for evaluation. The crevasses are located in the UAV imagery in the background, but the SAM and SAM 2 models are segmenting the glacier's ice in the foreground. Therefore, each segmentation mask output from the model must have the zero and one pixel values reversed to display the segmentation of the crevasses on the output mask. The converted segmentation masks output from the SAM and SAM 2 models were evaluated visually and with performance metrics for image segmentation. The DSC and Intersection over Union (IoU) metrics have been used because of the imbalance between the foreground and background pixel classes. There are more foreground pixels in the UAV imagery than in the background after the output segmentation masks have their zero and one values reversed.

## 4.2 Point Prompt Experiments

Further experiments were run with the SAM and SAM 2 models by applying a single background and foreground point prompt to the UAV images from Borebreen. The manual point prompt experiments were run using the single-mask and multi-mask settings on both models. The single-mask and multi-mask binary segmentation results were evaluated

visually and with performance metrics for image segmentation. For the same pixel class imbalance reasons as the mask generator module experiments, the DSC and IoU performance metrics were used. However, instead of reversing the zero and one pixel values in each segmentation mask output, the background (red) and foreground (green) point prompts were reversed. A single background (red) point prompt was placed on the glacier's ice, and a single foreground (green) point prompt was placed inside a crevasse in the UAV images evaluated. Placing the point prompts in this order prompts the SAM and SAM 2 models to segment the crevasses on the glacier without needing to reverse the output segmentation mask's zero and one pixel class values after modelling.

## 5 Results & Discussion

### 5.1 Mask Generator Prompting

After running tests with the SAM and SAM 2 mask generator modules, the automated prompting performance was evaluated on the ten UAV images. Ten test images have been used because of how labour-intensive and time-consuming it is to annotate images and gain expert approval. In the time frame for writing the paper, all the labelled images for semantic segmentation have been used in the experiments. However, ten 1024 x 1024 resolution UAV images are the same as having 160, 256 x 256 images which is a common resolution used for deep learning computer vision experiments. The performance evaluation results are included in Table 1, which show that the SAM 2 Hiera-L model outperforms the other SAM models with average DSC and IoU scores of 0.43 and 0.28, respectively. However, when inspecting the segmentation masks from each model, it was noticed that both models detected false positives around the areas of the segmented crevasses. The

walls inside some of the crevasses were found to have pixel values similar to those of the glacier's surface, which the SAM models are mistaken for crevasses. The SAM models were also noted to detect false negatives because the dark pixel colours inside the crevasses are the same colour as the sediment that grows on the glacier surface outside the crevasses. The SAM 2 Hiera-L model outperformed the SAM ViT-H model because the visual results display that they align with the SAM 2 Hiera-L model's average DSC and IoU performance metrics in Table 1. Although the SAM 2 Hiera-L model has a smaller, more efficient architecture than the SAM ViT-H model, the results display how the largest SAM 2 model can outperform larger models with more parameters. The visual segmentation results for the SAM and SAM 2 model mask generator outputs are included in Figures A.1. to A.20 in the appendix.

The average DSC and IoU scores were analysed with a line plot that includes a dust line to display the average DSC and IoU values and the overall average DSC and IoU values for the complete range between models in Figure 3. The line plots were used to visually display the DSC and IoU model performance metrics against one another and their data distributions for the ten UAV images in the test dataset. The dust line plots show that the SAM 2 Hiera-L performs the best and has a data distribution over a smaller range than the other SAM models except for image 6. Image 6 is an outlier because it is very challenging for the SAM models to segment crevasses because of how small the crevasses are in the image compared to the other nine images evaluated. Therefore, all of the performance evaluation results and plots indicate that the SAM 2 Hiera-L model performs the best because it has been trained on the largest pre-training dataset and has the largest model architecture of the SAM 2 models.

## 5.2 Point Prompting

After evaluating the performance of the automated prompting experiments using the SAM and SAM 2 mask generator modules, manual point prompt experiments were evaluated. The evaluation test results from the single-mask and the best performing multi-mask output for each SAM and SAM 2 model have been presented in Tables 2 and 3 respectively. Table 2 shows that the SAM ViT-H model outperforms the other SAM and SAM 2 models when applying point prompts in the single-mask mode with average DSC and IoU values of 0.15 and 0.08. However, after evaluating the best overall performance from the three segmentation masks output from the multi-mask mode for the SAM and SAM 2 models in Table 3, it was found that the SAM 2 Hiera-L and SAM ViT-H performs almost on par with the SAM ViT-L model. The SAM ViT-L performed the best

during multi-mask mode point prompt experiments with DSC and IoU values of 0.21 and 0.12. The dust line plots display the reduced DSC and IoU results over a larger data distribution range for the single and multi-mask experiments with the SAM and SAM 2 models over the mask generator results. The dust line plots are included in Figures A.21 and A.22 in the appendix. The SAM models generate confidence scores through a combination of mask prediction, refinement using attention mechanisms and the conversion of logits to probabilities using the sigmoid activation function. Therefore, the confidence scores generated in the multi-mask setting of the SAM models help guide the model to a better output perdition. A better output prediction is made because three masks are generated and scored to reflect the model's certainty about each mask generated instead of one in the single-mask mode. Using the multi-mask mode of the SAM models helps limit the adverse effects on the model from ambiguity, complex shapes, overlapping objects, and noisy data. Overall, the manual point prompt experiments with both single-mask and multi-mask modes set for the SAM and SAM 2 models did not outperform the segmentation results of the SAM 2 Hiera-L model using the mask generator module's automated prompting.

# 6    Conclusion & Future Work

All of the available SAM and SAM 2 models have been evaluated on ten UAV images from the Borebreen glacier. The SAM 2 Hiera-L model was found to perform better during the evaluation of this work for automated prompting experiments than the other SAM and SAM 2 models. During the automated prompting experiments, the SAM 2 Hiera-L model outperformed all SAM and SAM 2 models run with single and multi-mask point prompt tests. This displays that foundation models that are the size of the SAM 2 Hiera-L architecture and trained on the same quantity of image data has the potential to be used on challenging out-of-domain images for EO segmentation applications. However, as the DSC and IoU performance metrics are lower than required for glaciology experts to analyse the model outputs, further fine-tuning techniques are required to be implemented to increase model performance. Finetuning is required because of how far out of the domain the UAV images are from the images in the SA-1B and SA-V training datasets for the SAM and SAM 2 foundation models. In future work few-shot and weakly supervised learning are techniques that can be used for finetuning to overcome data shortages and labelling challenges. Depth estimation, image classification and object detection for crevasses could also be included in future works.

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

# A    SAM & SAM 2 Results

Section A includes the visual segmentation results from the SAM and SAM 2 model's automatic prompting experiments in Figures A.1 to A.20 and two dust line plots from the manual point prompting experiments using the single and multi mask modes in Figures A.21 to A.22 that commence over the page. The code is available at: https://github.com/Stevieee83/Exploring-Segment-Anything-Foundation-Models-for-Out-of-Domain-Crevasse-Drone-Image-Segmentation

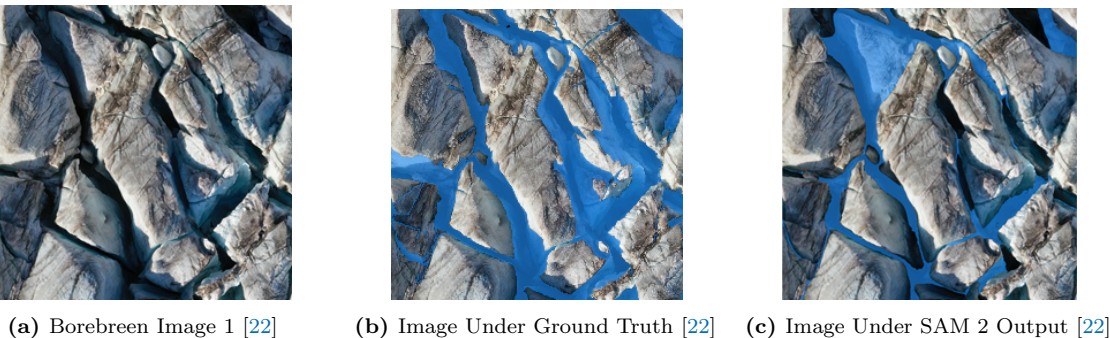

**(a)** Borebreen Image 1 [22]  **(b)** Image Under Ground Truth [22]  **(c)** Image Under SAM 2 Output [22]

**Figure A.1.** Borebreen Image 1 with the ground truth over the image and the SAM 2 model output over the same image. The background that has been segmented is blue in colour for the ground truth and the SAM 2 model output.

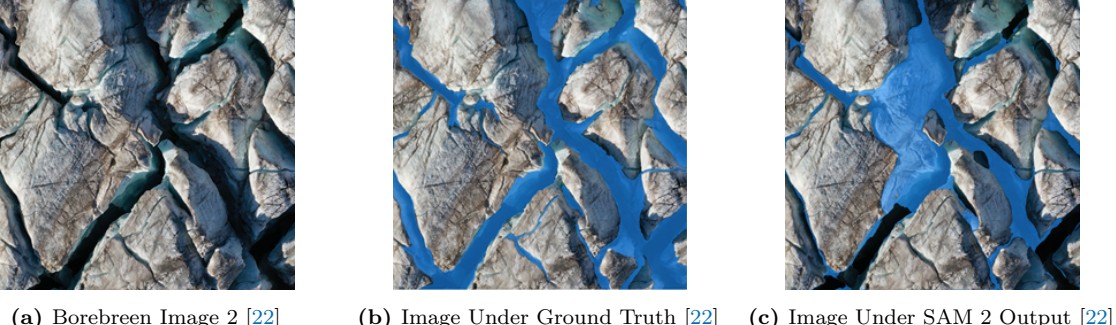

**(a)** Borebreen Image 2 [22]  **(b)** Image Under Ground Truth [22]  **(c)** Image Under SAM 2 Output [22]

**Figure A.2.** Borebreen Image 2 with the ground truth over the image and the SAM 2 model output over the same image. The background that has been segmented is blue in colour for the ground truth and the SAM 2 model output.

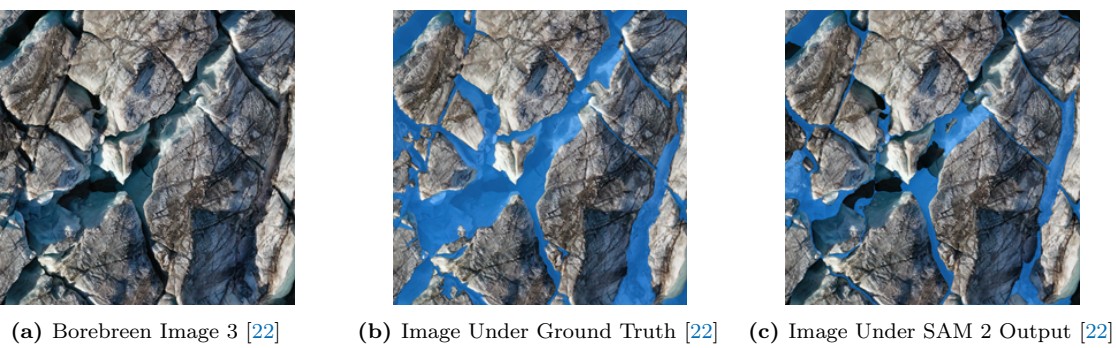

**(a)** Borebreen Image 3 [22]  **(b)** Image Under Ground Truth [22]  **(c)** Image Under SAM 2 Output [22]

**Figure A.3.** Borebreen Image 3 with the ground truth over the image and the SAM 2 model output over the same image. The background that has been segmented is blue in colour for the ground truth and the SAM 2 model output.

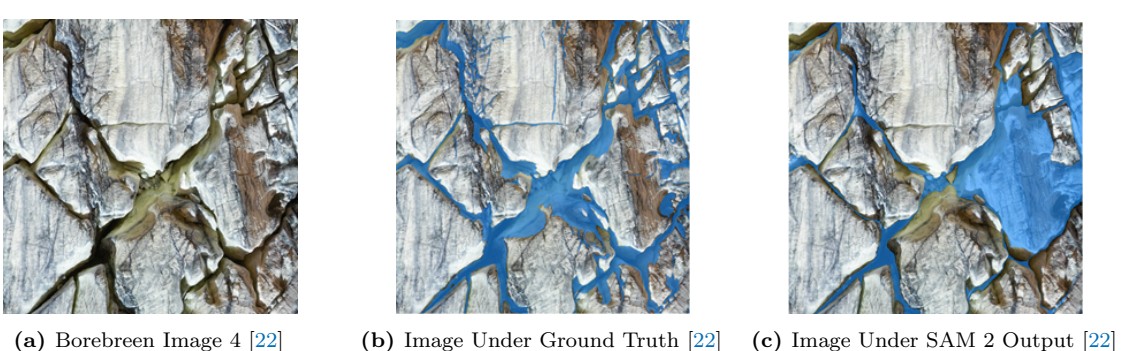

**(a)** Borebreen Image 4 [22]  **(b)** Image Under Ground Truth [22]  **(c)** Image Under SAM 2 Output [22]

**Figure A.4.** Borebreen Image 4 with the ground truth over the image and the SAM 2 model output over the same image. The background that has been segmented is blue in colour for the ground truth and the SAM 2 model output.

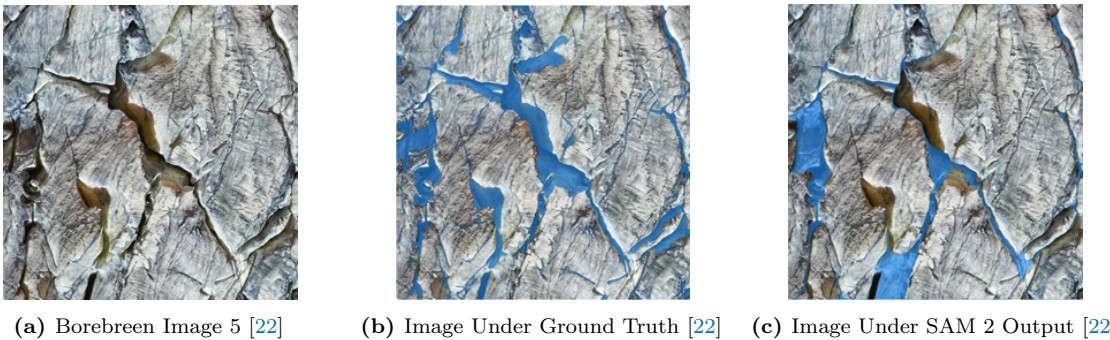

**(a)** Borebreen Image 5 [22]     **(b)** Image Under Ground Truth [22]     **(c)** Image Under SAM 2 Output [22]

**Figure A.5.** Borebreen Image 5 with the ground truth over the image and the SAM 2 model output over the same image. The background that has been segmented is blue in colour for the ground truth and the SAM 2 model output.

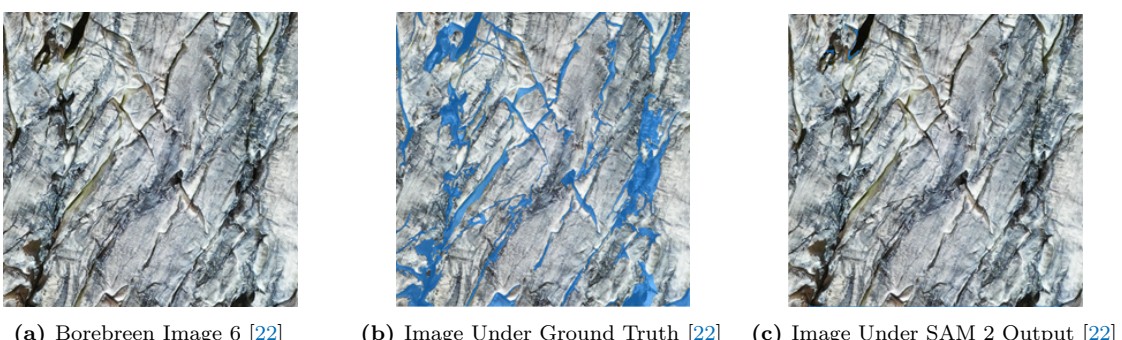

**(a)** Borebreen Image 6 [22]     **(b)** Image Under Ground Truth [22]     **(c)** Image Under SAM 2 Output [22]

**Figure A.6.** Borebreen Image 6 with the ground truth over the image and the SAM 2 model output over the same image. The background that has been segmented is blue in colour for the ground truth and the SAM 2 model output.

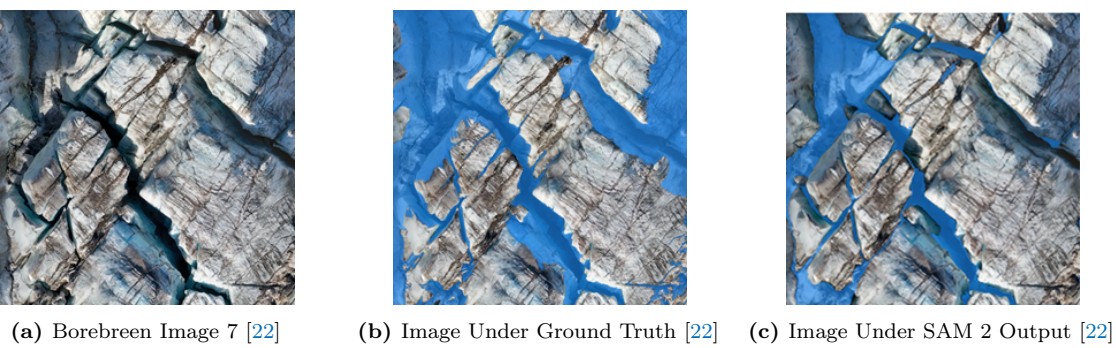

**(a)** Borebreen Image 7 [22]     **(b)** Image Under Ground Truth [22]     **(c)** Image Under SAM 2 Output [22]

**Figure A.7.** Borebreen Image 7 with the ground truth over the image and the SAM 2 model output over the same image. The background that has been segmented is blue in colour for the ground truth and the SAM 2 model output.

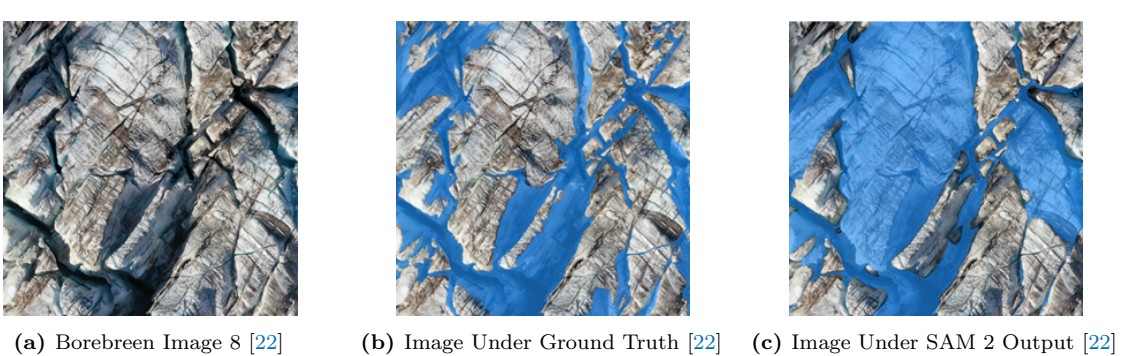

**(a)** Borebreen Image 8 [22]     **(b)** Image Under Ground Truth [22]     **(c)** Image Under SAM 2 Output [22]

**Figure A.8.** Borebreen Image 8 with the ground truth over the image and the SAM 2 model output over the same image. The background that has been segmented is blue in colour for the ground truth and the SAM 2 model output.

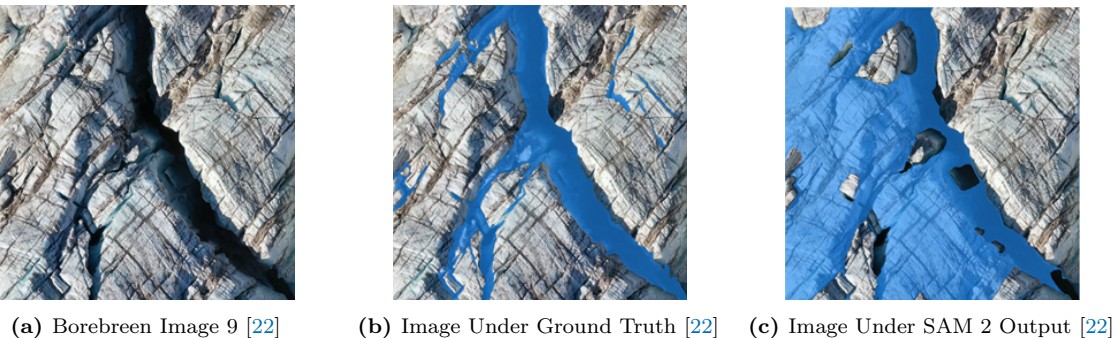

**(a)** Borebreen Image 9 [22]          **(b)** Image Under Ground Truth [22]          **(c)** Image Under SAM 2 Output [22]

**Figure A.9.** Borebreen Image 9 with the ground truth over the image and the SAM 2 model output over the same image. The background that has been segmented is blue in colour for the ground truth and the SAM 2 model output.

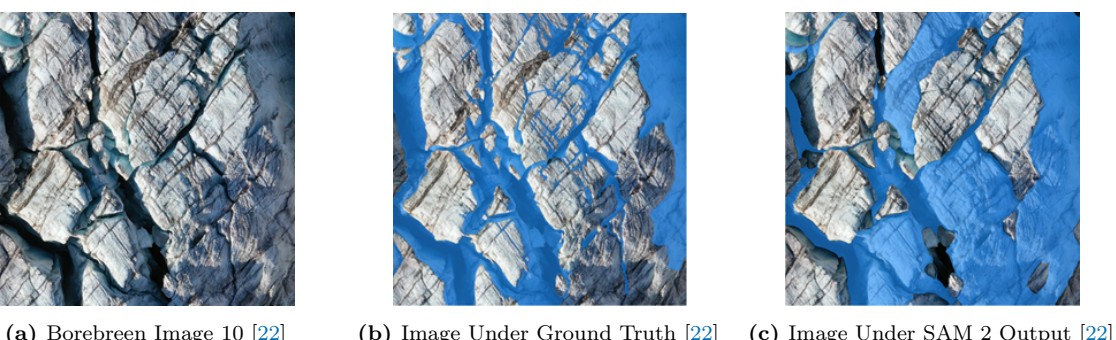

**(a)** Borebreen Image 10 [22]          **(b)** Image Under Ground Truth [22]          **(c)** Image Under SAM 2 Output [22]

**Figure A.10.** Borebreen Image 10 with the ground truth over the image and the SAM 2 model output over the same image. The background that has been segmented is blue in colour for the ground truth and the SAM 2 model output.

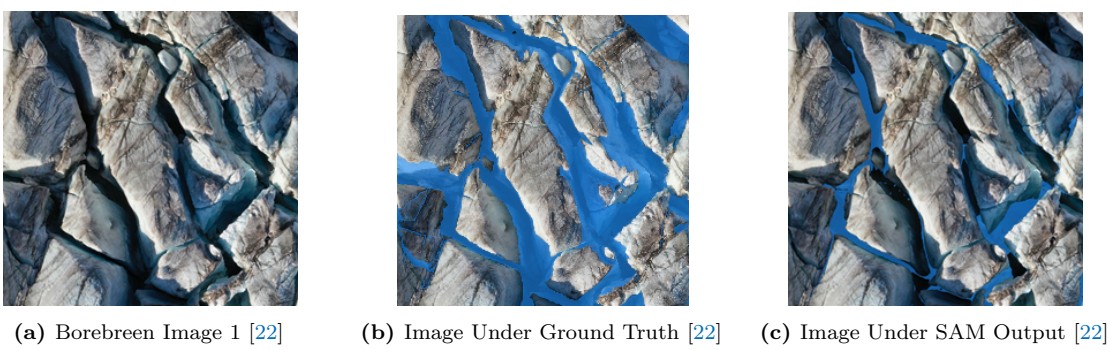

**(a)** Borebreen Image 1 [22]          **(b)** Image Under Ground Truth [22]          **(c)** Image Under SAM Output [22]

**Figure A.11.** Borebreen Image 1 with the ground truth over the image and the SAM model output over the same image. The background that has been segmented is blue in colour for the ground truth and the SAM model output.

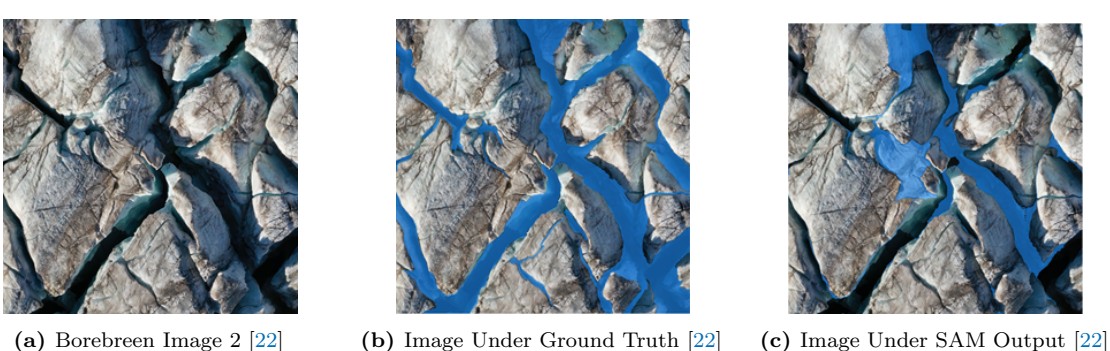

**(a)** Borebreen Image 2 [22]          **(b)** Image Under Ground Truth [22]          **(c)** Image Under SAM Output [22]

**Figure A.12.** Borebreen Image 2 with the ground truth over the image and the SAM model output over the same image. The background that has been segmented is blue in colour for the ground truth and the SAM model output.

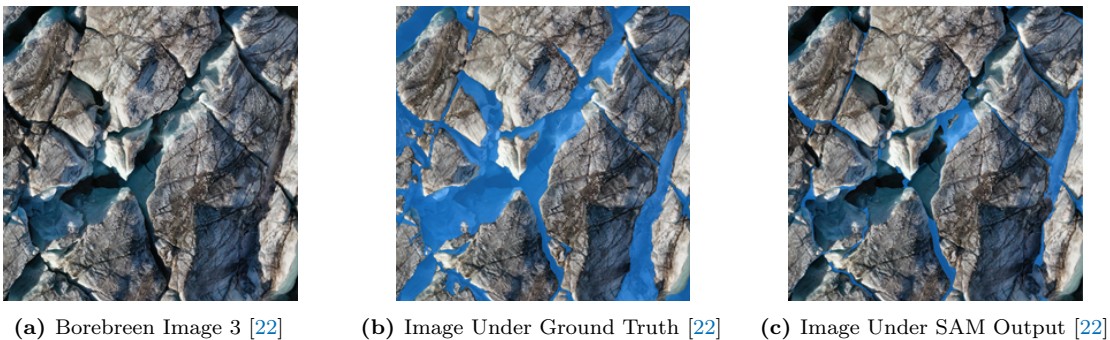

**(a)** Borebreen Image 3 [22]    **(b)** Image Under Ground Truth [22]    **(c)** Image Under SAM Output [22]

**Figure A.13.** Borebreen Image 3 with the ground truth over the image and the SAM model output over the same image. The background that has been segmented is blue in colour for the ground truth and the SAM model output.

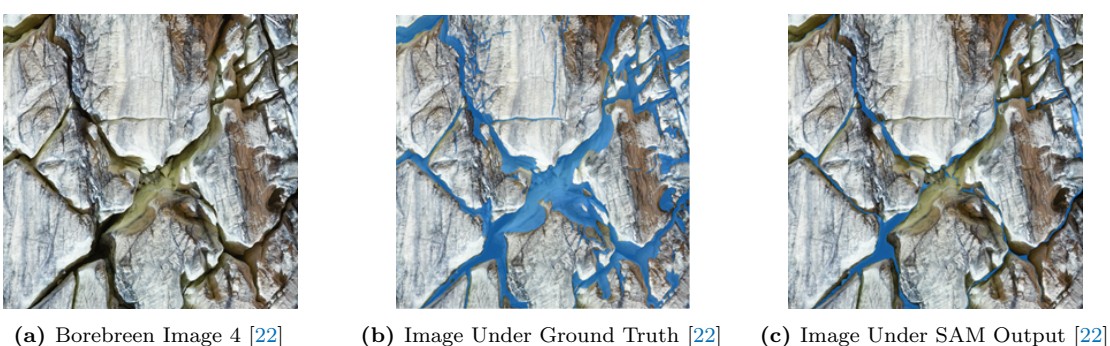

**(a)** Borebreen Image 4 [22]    **(b)** Image Under Ground Truth [22]    **(c)** Image Under SAM Output [22]

**Figure A.14.** Borebreen Image 4 with the ground truth over the image and the SAM model output over the same image. The background that has been segmented is blue in colour for the ground truth and the SAM model output.

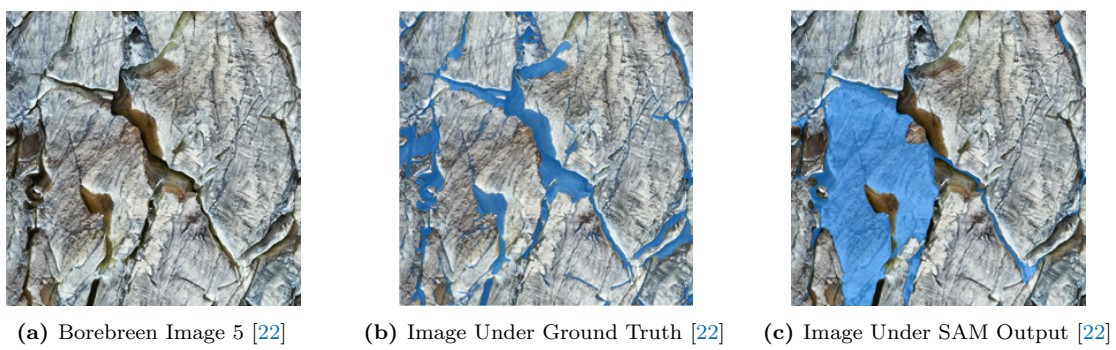

**(a)** Borebreen Image 5 [22]    **(b)** Image Under Ground Truth [22]    **(c)** Image Under SAM Output [22]

**Figure A.15.** Borebreen Image 5 with the ground truth over the image and the SAM model output over the same image. The background that has been segmented is blue in colour for the ground truth and the SAM model output.

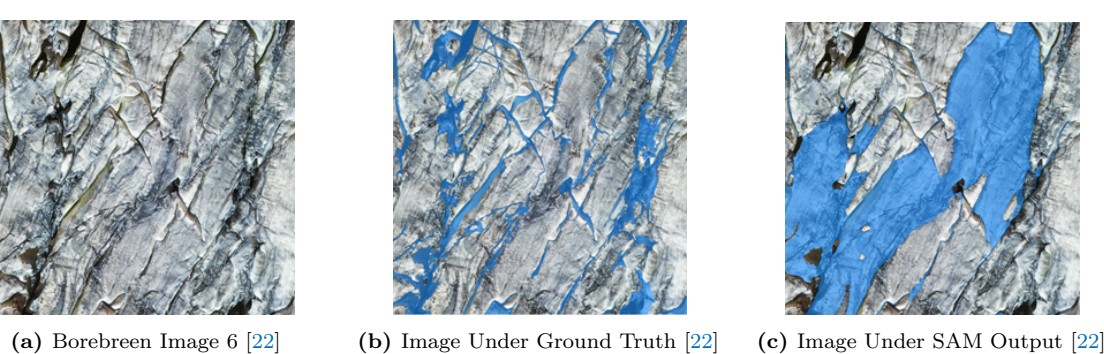

**(a)** Borebreen Image 6 [22]    **(b)** Image Under Ground Truth [22]    **(c)** Image Under SAM Output [22]

**Figure A.16.** Borebreen Image 6 with the ground truth over the image and the SAM model output over the same image. The background that has been segmented is blue in colour for the ground truth and the SAM model output.

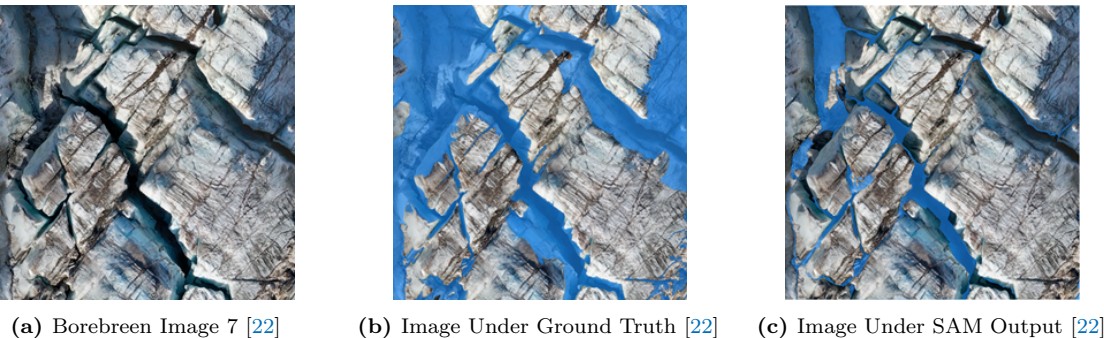

**(a)** Borebreen Image 7 [22]  **(b)** Image Under Ground Truth [22]  **(c)** Image Under SAM Output [22]

**Figure A.17.** Borebreen Image 7 with the ground truth over the image and the SAM model output over the same image. The background that has been segmented is blue in colour for the ground truth and the SAM model output.

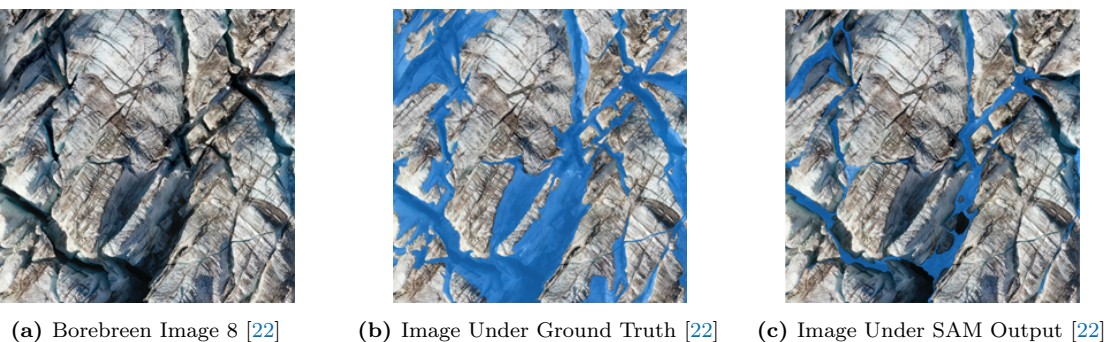

**(a)** Borebreen Image 8 [22]  **(b)** Image Under Ground Truth [22]  **(c)** Image Under SAM Output [22]

**Figure A.18.** Borebreen Image 8 with the ground truth over the image and the SAM model output over the same image. The background that has been segmented is blue in colour for the ground truth and the SAM model output.

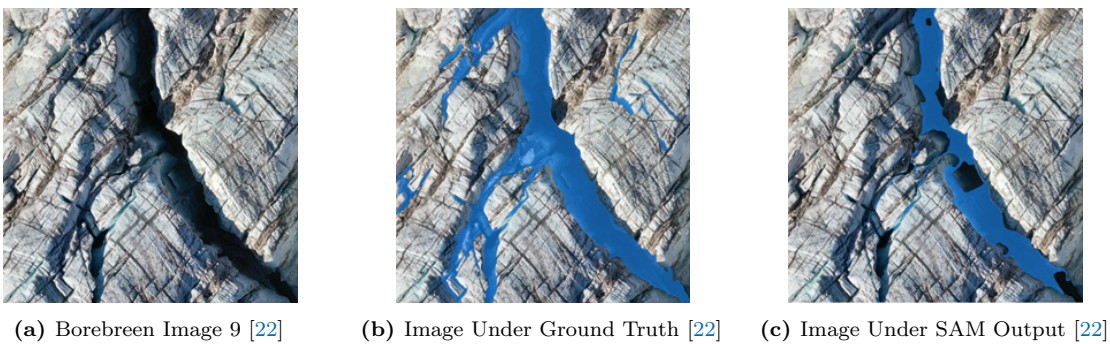

**(a)** Borebreen Image 9 [22]  **(b)** Image Under Ground Truth [22]  **(c)** Image Under SAM Output [22]

**Figure A.19.** Borebreen Image 9 with the ground truth over the image and the SAM model output over the same image. The background that has been segmented is blue in colour for the ground truth and the SAM model output.

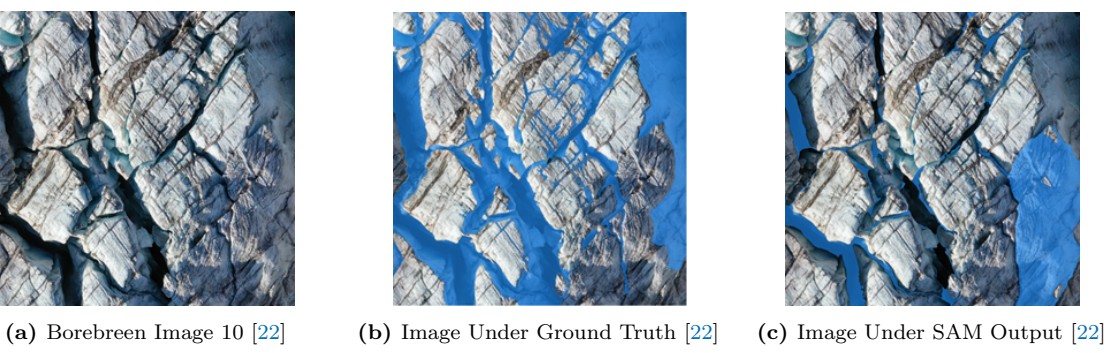

**(a)** Borebreen Image 10 [22]  **(b)** Image Under Ground Truth [22]  **(c)** Image Under SAM Output [22]

**Figure A.20.** Borebreen Image 10 with the ground truth over the image and the SAM model output over the same image. The background that has been segmented is blue in colour for the ground truth and the SAM model output.

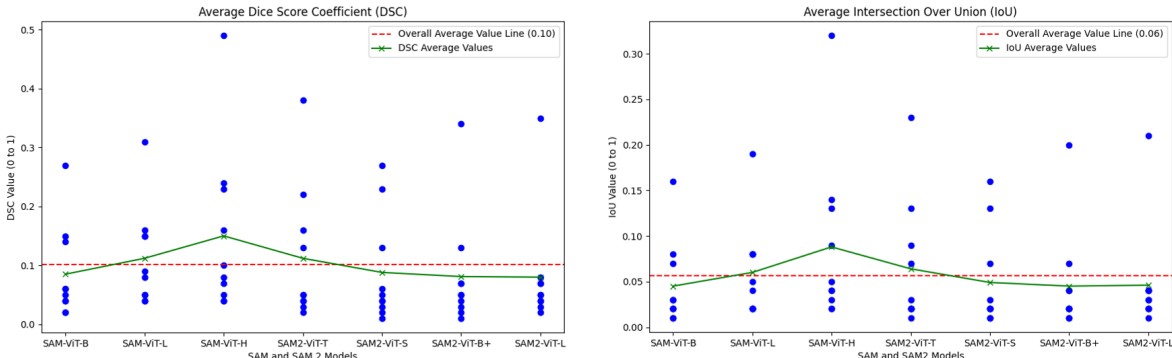

**Figure A.21.** DSC and IoU dust line plots for the SAM and SAM 2 results on all seven models for single-mask point prompt experiments.

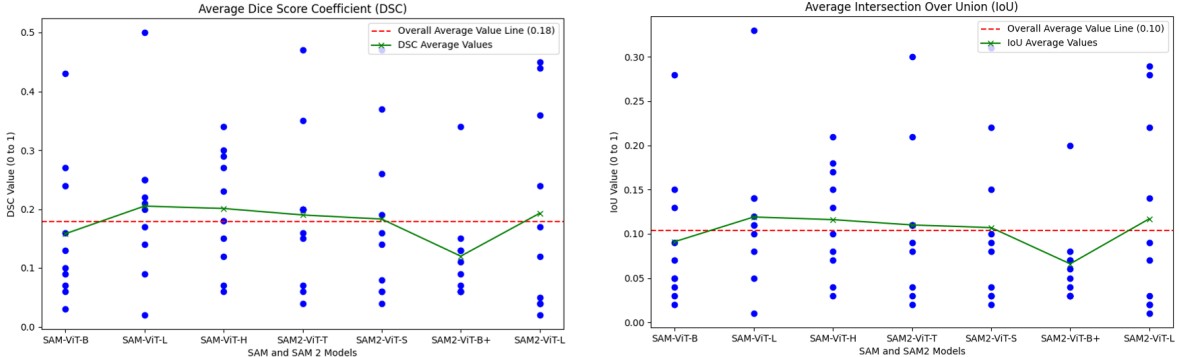

**Figure A.22.** DSC and IoU dust line plots for the SAM and SAM 2 results on all seven models for multi-mask point prompt experiments.

