# OpenReview forum: "Exploring Segment Anything Foundation Models for Out of Domain Crevasse Drone Image Segmentation"
_NLDL.org/2025/Conference — NLDL 2025 Oral_

### Official Review · Reviewer_YfWd · 2024-09-25
**Application of Segment Anything (foundation) models for detecting crevasses in UAV imagery**

**Confidence:** 3

**Summary:**

The paper aims to quantify the generalization ability of the foundation models known as SAM (Segment anything Model) and SAM 2 on UAV imagery of arctic glaciers for the purpose of detecting crevasses.
Without any baselines, the paper compares several variants of SAM and SAM2 and employs the Dice score (DSC) and intersection-over-union (IOU) as metrics for performance. The paper concludes that the best-performing model is not good enough for downstream applications and fine-tuning is required.

**Strengths:**

The paper addresses an important and well-motivated problem.
Overall, the paper is well-written, well-structured, and easy to follow.

**Weaknesses:**

The only novelty of the paper is the application of SAM and SAM2 to UAV data, but the paper provides very little details about the applied methodology. Despite the use of off-the-shelf models, it seems like it would be very hard to reproduce the results. There are no details about how each of the models, prompts, etc. are configured.

Moreover, the evaluation of the model is also quite weak. First of all, there are no baseline models included as a reference. If the authors had provided the performance metrics for simple baselines as predicting the majority class or random guessing or a stronger baseline such as a trained U-NET (even for the small dataset they have) would help put the reported performance metrics into perspective. The simple baselines are really relevant since several of the models seem to perform very poorly (i.e. IoC as low as 0.03).

The choice of metrics also seems a bit odd. The paper uses IoU and DSC, which are essentially quantifying the same thing (you can write one as a function of the other). The similarity of the metrics is also very evident from Figure 3. It would have been much more informative to include other metrics or use the space for elaborating on the methodology. Finally, it is not clear to me what the purpose of the analysis of Figure 3.

Minor details: The abbreviation 'DSC' does not seem to be defined, but I assume it is the Dice score.

**Final Rebuttal Confidence:**

4

**Final Rebuttal Justification:**

Thank you for the rebuttal. However, I am maintaining my score because of the weak experimental evaluation as my concern about basics, such as proper metrics, relevant baselines etc. must be included to help put the results in perspective and therefore baselines cannot be left for future work.

**Justification:**

Since the sole contribution of the paper is the application and evaluation of SAM models on a new type of data, it is problematic that the methodology is carefully described and the evaluation is weak (no baseline methods included, using IoC and DSC which are heavily correlated).

---

> ### Author Rebuttal · Authors · 2024-10-25
>
> We would like to take this opportunity to thank you for your review of our paper and hope that we can alleviate your concerns below.
>
> **The only novelty of the paper is the application of SAM and SAM2 to UAV data, but the paper provides very little details about the applied methodology. Despite the use of off-the-shelf models, it seems like it would be very hard to reproduce the results. There are no details about how each of the models, prompts, etc. are configured.**
>
> We have updated section the Methods section 4 to explain to the reader that the code for the mask generator module is in the SAM and SAM 2 official GitHub repository and that it was used for each model to generate prompts automatically.
>
> **Moreover, the evaluation of the model is also quite weak. First of all, there are no baseline models included as a reference. If the authors had provided the performance metrics for simple baselines as  predicting the majority class or random guessing or a stronger baseline  such as a trained U-NET (even for the small dataset they have) would  help put the reported performance metrics into perspective. The simple baselines are really relevant since several of the models seem to perform very poorly (i.e. IoC as low as 0.03).**
>
> The experiments carried out on the SAM and SAM 2 models used inference mode to evaluate and compare the SAM and SAM 2 models against one another with no additional fine-tuning. A test dataset of five 1024 x 1024 images was used in the original paper as that was all of the labelled images available at the time because of how labour-intensive and time-consuming it is to label the image data and seek expert approval for crevasse segmentation. The test dataset has been increased to 10, 1024 x 1024 images (equivalent to 160 256 x 256 resolution images) because more images have been labelled since the original submissoin of the paper. Weakly supervised and few-shot learning will be explored in future works, as explained in the Conclusion and Future Work section 6 of the manuscript and can be used to benchmark against other models such as U-Net, DeepLabV3+ and other foundation/segmentation models.
>
> **The choice of metrics also seems a bit odd. The paper uses IoU and DSC, which are essentially quantifying the same thing (you can write one as a function of the other). The similarity of the metrics is also very evident from Figure 3. It would have been much more informative to include other metrics or use the space for elaborating on the methodology. Finally, it is not clear to me what the purpose of the analysis of Figure 3.**
>
> The DSC and IoU metrics have been used to compare the paper against segmentation studies that use one metric over the other. Therefore, the paper will be comparable to more studies. DSC and IoU have been used because the UAV drone images for crevasse segmentation contain a higher number of foreground pixels as opposed to background pixels. Therefore, the UAV image data has a heavily imbalanced number of examples (pixels) between classes where a high level of accuracy, precision or recall could be achieved for a model that performs below the standard for image segmentation. Using DSC and IoU fairly evaluates the models and displays a more accurate representation than other performance metrics for image segmentation.
>
> **Minor details: The abbreviation 'DSC' does not seem to be defined, but I assume it is the Dice score.**
>
> The term DSC has been updated in the main body of the manuscript to inform the reader it is the Dice Score Coefficient.

---

### Official Review · Reviewer_X91q · 2024-09-27
**Good exploration of target problem domain but experimentally weak**

**Confidence:** 4

**Summary:**

The paper describes the problem of segmenting images of glaciers in the Arctic to identify glaciers. Unlike glaciers in the Antarctic, those in the Arctic are challenging to segment because the crevasses are smaller.

Segment Anything Model (SAM) 1 & 2 are applied in a zero-shot fashion to the problem. It is found that SAM 1 & 2 are confounded by sediment and areas around the crevasse, and the performance of the approach is inadequate as a replacement for manual labelling.

Finally, the paper recommends that some form of fine-tuning is required to tackle the problem to the point that the results of the AI model applied would be useful.

**Strengths:**

- The paper clearly demonstrates a deep level of understanding of domain-specific glacial mechanics, and weaknesses of the described approach are clearly and concisely described
- Reflection on the methodology used is clear and detailed
- Some novelty is shown through an incremental contribution is made to the problem outlined over a previous approach
- The problem described is difficult and non-trivial, citing challenges about sourcing labelled data.
- An innovative solution to the problem is presented, making use of the latest advancements in the field of computer vision.
- The paper makes the problem and the proposed solution clear and easy to understand
- No ethical issues are obviously present
- Experiments done are thoroughly analysed given the circumstances
- Sources are clearly referenced for all ideas described.
- No conflicts of interest or undisclosed affiliations could be found, but given the paper is anonymised it is recommended that the chairs analyse the non-anonymised paper for this.

**Weaknesses:**

- No link to source code or a data repository appears to have been provided.
	- It is suggested that all code written is uploaded to a git repository, and data is somehow freely available for other researchers to download
- It is unclear why the proposed future work of implementing a fine-tuning based solution was not implemented by the researchers.
- Limited insight is shown comparing the approach demonstrated to prior models, as no direct comparison between prior approaches and the proposed solution are given for the target dataset.
- The size of the target dataset is extremely limited
- No thought is given to alternative problem task formulations, such as depth estimation, classification, or object detection.
- Limited significance to the deep learning community is demonstrated
- The multi-mask prediction mode is described on lines 348/349, and an improvement in experimental performance is reported, but no metrics are shown and it is unclear if the experiments described contain this improvement or not.

**Justification:**

The researchers clearly demonstrate significant understanding of the target problem domain. The problem described is challenging. The paper displays some incremental novelty in improving on a prior approach.

However, the paper is experimentally weak. The presented methodology is not directly compared against prior approaches on the target dataset (which is extremely limited in size), even though it appears they are directly comparable. It is also unclear why the researchers did not implement their own suggestion of fine-tuning.

However, despite this the paper is clear, and accurately summarises prior work done on the problem by others in the field. It characterises the performance of SAM 1 & 2 on the target problem domain well, providing some useful insights to direct future research to solve the issue.

---

> ### Author Rebuttal · Authors · 2024-10-25
>
> We would like to take this opportunity to thank you for your review of our paper and hope that we can alleviate your concerns below.
>
> **No link to source code or a data repository appears to have been provided.**
>
> **It is suggested that all code written is uploaded to a git repository, and data is somehow freely available for other researchers to download**
>
> All material to reproduce the experiments will be made available after the review period if the manuscript is accepted.
>
> **It is unclear why the proposed future work of implementing a fine-tuning based solution was not implemented by the researchers.**
>
> The proposed fine-tuning methods have not been implemented because of the shortage of labelled UAV image data. After more UAV images have been labelled for segmentation, weakly supervised and few-shot learning will be carried out in future works, as documented in the Conclusion and Future Work section 6 of the manuscript.
>
> **Limited insight is shown comparing the approach demonstrated to prior models, as no direct comparison between prior approaches and the proposed solution are given for the target dataset.**
>
> The paper focuses on the evaluation of the SAM and SAM 2 models. As there is limited labelled data for segmentation, only a test dataset of ten images could be used for the experiments. Once more data is labelled, a comparison between existing image segmentation models for semantic segmentation will be carried out in future works.
>
> **The size of the target dataset is extremely limited**
>
> The size of the target dataset has been increased to ten images, and all of the experiments have been redone on the newly labelled images. At the time of originally writing the paper, five 1024 x 1024 images labelled for segmentation were all that was available because labelling images is time-consuming and labour-intensive. The new test dataset size of 10 1024 x 1024 images is the equivalent of 160 256 x 256 images commonly used for deep learning image segmentation experiments.
>
> **No thought is given to alternative problem task formulations, such as depth estimation, classification, or object detection.**
>
> Depth estimation, image classification, or object detection applications for crevasses have been added to the Conclusion and Future Work section 6 to be included in future works.
>
> **Limited significance to the deep learning community is demonstrated**
>
> The manuscript's Motivation and Challenges subsection 1.1 includes further information on how the SAM 2 model could be used to start the labelling process for semantic segmentation to reduce the time and labour required.
>
> **The multi-mask prediction mode is described on lines 348/349, and an improvement in experimental performance is reported, but no metrics are shown and it is unclear if the experiments described contain this improvement or not**
>
> A link has been provided to Tables 1, 2 and 3 in the manuscript that takes the readers to the results table they would like to view. Tables 1, 2 and 3 contain the results from all the experiments (Mask Generator and Point Prompt tests in single-mask and multi-mask modes). The best average DCE and IoU metric values for the Mask Generator and Multi-Mask point prompt experiments have been added to the text in the Results and Discussion section 5 of the manuscript as recommended.

---

### Official Review · Reviewer_LSff · 2024-10-02
**A well-written manusciprt**

**Confidence:** 3

**Summary:**

This paper studies drone images of crevasses in Arctic, which is an important but less explored domain. Authors apply SAM and SAM models to segment crevasses and qualitively evaluated the resutls, which suggests that further fine-tuning is still essential for using machine learning models on the crevasses segmentation task.

**Strengths:**

- The problem of segmenting crevasses in Arctic is an important issue by itself.
- Authors take advantage of the most advanced segmentation foundation models.
- The paper is overall well written and easy to follow. Authors summarized relate research in a clear way. The evaluation results is also clearly presented.

**Weaknesses:**

- It seems like the full power of SAM models are not fully elicited.
  - According to section 4, authors use SAM models to conduct inference in two situations: 1. grid points prompt 2. two single points prompt of background and foreground.
  - According to Figure 2, grid points for the images may result in point being located in both the crevasse region and non-crevasse region, which will make the segmentation results worse.
  - Considering the image resolution and the distribution of crevasses in the images, it woule be neccessary to adjust the sparsity of the grid points prompts.

- For figure 2, descriptions are missing why the resolution 1024x1024 is chosen. The ground truth of crevasse has quite complex shapes in the current resolution. Considering SA-1B and SA-V datasets are mostly about common objects, decress the dron image resolution to have simpler shape crevasses may help.

- (Please forgive me on lacking relevent knowledge.) Five images seem like too few for a valid evalution. Difficulites on collecting data (like cost and time span) is not described in the manuscript.

- Further discussion on using SAM models to relieve the burden of annotation can be appended.

**Final Rebuttal Confidence:**

3

**Final Rebuttal Justification:**

My concerns on experiment settings are resolved, and the size of the dataset is not a main issue due to the difficulty of collecting these dsata. The quantitative results is also very inclusive.
However, the utilization of SAM models is insufficient and page limit is not a relevant reason. I encourage authors to explore and report more details on the results when adjusting the models, including image sizes, image conditions (such as the density of crevasses shown in the image), number of prompt points, format of prompts (such as words or bounding boxes), internal parameters of the models (such as thresholds).

**Justification:**

Lacking background knowledge on the glacier field, I gave an assessment from the machine learning point of view.
- Dataset size is too small.
- The effectiveness of the foundation models are not thoroughly examined. There are potentials to improve from both the data side and the model side.

---

> ### Author Rebuttal · Authors · 2024-10-25
>
> We would like to take this opportunity to thank you for your review of our paper and hope that we can alleviate your concerns below.
>
> **It seems like the full power of SAM models are not fully elicited.**
>
> Thank you for your feedback. Various settings in the mask generator module were used, and the setting that produced the best results was used to record the segmentation results for the manuscript and then applied across both the SAM and SAM 2 models for fair evaluation.
>
> **According to section 4, authors use SAM models to conduct inference in two situations: 1. grid points prompt 2. two single points prompt of background and foreground.**
>
> We have updated the manuscript in the Methods section 4 to include more detail on implementing the experiments. The experiments were run with SAM and SAM 2 using the Mask Generator module, where code is available in the GitHub repository for each model. Two further methods were used, with point prompting available to SAM and SAM 2. Both the single-mask and multi-mask modes were used, and the best results for each ViT model for SAM and SAM 2 have been recorded in Tables 2 and 3 of the manuscript on ten 1024 x 1024 UAV images.
>
> **According to Figure 2, grid points for the images may result in point being located in both the crevasse region and non-crevasse region, which will make the segmentation results worse.**
>
> The information in Figure 2 has been made clear with additional information and links provided in the text to where the results apply to in the Results and Discussion section.
>
> **Considering the image resolution and the distribution of crevasses in the images, it woule be neccessary to adjust the sparsity of the grid points prompts.**
>
> We have updated the manuscript to include information on why an image resolution of 1024 x 1024 was used for the experiments in the UAV Data section 3. An image resolution of 1024 x 1024 has been used because the SAM model will interpolate smaller images in resolution i.e. 256 x 256 or 512 x 512 up to 1024 x 1024 to align with the input dimensions of the ViT-B, ViT-L and ViT-H image encoders available for the model. Therefore, as interpolation decreases the quality of the images input to the model, the segmentation performance will decrease unfairly for the SAM model using interpolation for super-resolution.
>
> **(Please forgive me on lacking relevent knowledge.) Five images seem like too few for a valid evalution. Difficulites on collecting data (like cost and time span) is not described in the manuscript.**
>
> Thank you for advising that the test dataset is very small. Five 1024 x 1024 images were all the annotated images available for segmentation at the time of writing the paper. A further 5 1024 x 1024 images have been annotated since the paper was originally written. The experiments were repeated on the larger test dataset of 10 images, and the results tables and figures were updated. The manuscript has been updated in section 3 UAV Data to explain why ten 1024 x 1024 images have been used for the experiments.
>
> **Further discussion on using SAM models to relieve the burden of annotation can be appended.**
>
> We have moved the information on using the SAM models to relieve the burden of annotation from the Methods section 4 to the Motivation and Challenges subsection 1.1 with further information as it is more suited to this section of the manuscript.

---

### Official Review · Reviewer_fd4r · 2024-10-09
**Easy-to-follow paper on segmentation in glaciology domain**

**Confidence:** 4

**Summary:**

Authors describe testing pre-trained models on crevasse segmentation in glaciology domain using UAV images of glaciers. One expert glaciologist visually evaluated segmentation results from seven models for the automatic mask generator prompt experiments. Authors report that SAM 2 Hiera-L model performs the best because it has been trained on the largest pre-training dataset and has the largest model architecture of the SAM 2 models. Authors argue that fine-tuning is required because of how far out of the domain the UAV images are from the images in the SA-1B and SA-V training datasets for the SAM and SAM 2 foundation models

**Strengths:**

Paper is easy to follow and I personally like the evaluation details presented as graphs. The paper could provide significant impact on glaciology

**Weaknesses:**

1. I would suggest spending more time on writing introduction and motivation with many more appropriate references since few strong statements lack references.
2. How does the problem of segmenting crevasses relate to broader segmentation challenges in remote sensing?
3. I believe authors lack discussion on what specific attributes of UAV images make them significantly different from the training datasets?
4. Authors did not investigate investigate (or report) whether conditions (e.g., lighting, glacier surface conditions, weather condition) affect false positive or false negative rates? Could additional pre-processing steps mitigate these errors?
5. Could the insights gained from other studies be applicable to this domain (e.g., agriculture, urban planning, etc)?
6. How would authors justify the balance between automated and manual prompting?

**Justification:**

I judge the paper as worthy of being shared with the community because it addresses issues in glaciology domain which might be under-represented in papers of similar conferences. I would love to see more abstract discussions stepping away from statistics and models applied. Similar to all data processing and analysis work, I wish authors would spend more time on talking about the data and show deeper understanding of the data from this domain. Perhaps access to the expert glaciologist could be a good start

---

> ### Author Rebuttal · Authors · 2024-10-25
>
> We would like to take this opportunity to thank you for your review of our paper and hope that we can alleviate your concerns below.
>
> **I would suggest spending more time on writing introduction and motivation with many more appropriate references since few strong statements lack references**
>
> The introduction and motivation sections have been updated as recommended, with more references to back up our claims.
>
> **How does the problem of segmenting crevasses relate to broader segmentation challenges in remote sensing?**
>
> The challenges around segmentation have been included in the Motivations and Challenges section of the paper, which we have revised in line with your recommendations.
>
> **I believe authors lack discussion on what specific attributes of UAV images make them significantly different from the training datasets?**
>
> The UAV images differ from the SA-1B and SA-V datasets because remote sensing images are classed as specialist images similar to images from the medical domain. A test dataset of 10 1024 x 1024 UAV images which is the equivallent to 160 256 x256 images have been used to evaluate and compare the SAM and SAM 2 pre-trained models in inference mode only.
>
> **Authors did not investigate investigate (or report) whether conditions (e.g., lighting, glacier surface conditions, weather  condition) affect false positive or false negative rates? Could additional preprocessing steps mitigate these errors?**
>
> An explanation that includes the weather conditions in the Arctic and why UAV drone data is used over satellite imagery from space has been included in section 3 UAV Drone Data. The UAV drones fly close to the ground, and the image data was captured in the summer months when there are 24 hours of daylight and less snow cover in the Arctic. UAV drones also fly below the cloud level. Therefore, unlike satellite imagery, cloud cover that prevents a clear image of the glacier from being captured is eliminated when using UAVs.
>
> Digital filter algorithms have been considered a preprocessing step, but we are trying to determine if a deep learning foundation model can segment crevasses without any preprocessing steps first.
>
> **Could the insights gained from other studies be applicable to this domain (e.g., agriculture, urban planning, etc)?**
>
> Other insights could be gained from agriculture and urban planning, but because glaciers have particular dynamics and features, our area of interest is in this area first.
>
> **How would authors justify the balance between automated and manual prompting?**
>
> The conclusions section has been updated to explain to the reader that the automated prompting with the mask generator module with the SAM 2 Heira-L performs the best over all other automated and manual point prompt tests.

---

### Meta-Review · Area_Chair_ek4J · 2024-10-31

**Recommendation:** Accept (Poster)
**Confidence:** 3

**Metareview:**

Overall, the authors have successfully improved the paper by incorporating the reviewers' feedback, enhancing its clarity for the reader. Based on the reviewers' comments on the authors' rebuttal and my own assessment of the paper, I recommend accepting this paper as a poster presentation.

**Suggested Changes To The Recommendation:**

2: I'm certain of the recommendation.  It should not be changed

---

### Decision · Program_Chairs · 2024-11-06

**Decision:**

Accept (Oral)

**Comment:**

We have decided to offer opportunities for oral presentations in the remaining available slots in the NLDL program. Thus, despite the AC's poster recommendation, we recommend an oral presentation in addition to the poster presentation given the AC's and reviewers' recommendations.